# Phase Angle and Ultrasound Assessment of the Rectus Femoris for Predicting Malnutrition and Sarcopenia in Patients with Esophagogastric Cancer: A Cross-Sectional Pilot Study

**DOI:** 10.3390/nu17010091

**Published:** 2024-12-29

**Authors:** Erika Vieira Maroun, María Argente Pla, María José Pedraza Serrano, Bianca Tabita Muresan, Agustín Ramos Prol, Eva Gascó Santana, Silvia Martín Sanchis, Ángela Durá De Miguel, Andrea Micó García, Anna Cebrián Vázquez, Alba Durbá Lacruz, Juan Francisco Merino-Torres

**Affiliations:** 1Joint Research Unit on Endocrinology, Nutrition and Clinical Dietetics, Health Research Institute La Fe, 46026 Valencia, Spain; erika.vieira@universidadeuropea.es (E.V.M.); merino_jfr@gva.es (J.F.M.-T.); 2Department of Medicine, Faculty of Medicine, University of Valencia, 46010 Valencia, Spain; 3Facultad Ciencias de la Salud, Universidad Europea de Valencia, 46010 Valencia, Spain; biancatabita.muresan@universidadeuropea.es; 4Endocrinology and Nutrition Department, La Fe University and Polytechnic Hospital in Valencia, 46026 Valencia, Spain; ramos_agu@gva.es (A.R.P.); gasco_evasan@gva.es (E.G.S.); martin_silsan@gva.es (S.M.S.); dura_angdem@gva.es (Á.D.D.M.); mico_and@gva.es (A.M.G.); cebrian_anavaz@gva.es (A.C.V.); durba_alb@gva.es (A.D.L.); 5Independent Researcher, 46100 Valencia, Spain; majopese@alumni.uv.es

**Keywords:** esophagogastric cancer, malnutrition, sarcopenia, ultrasound of rectus femoris muscle, phase angle, morphofunctional assessment

## Abstract

Background: Disease-related malnutrition (DRM) and sarcopenia are prevalent conditions in gastrointestinal cancer patients, whose early diagnosis is essential to establish a nutritional treatment that contributes to optimizing adverse outcomes and improving prognosis. Phase angle (PhA) and rectus femoris ultrasound measurements are considered effort-independent markers of muscle wasting, which remains unrecognized in oncology patients. Objective: This study aimed to evaluate the potential utility of PhA, rectus femoris cross-sectional area (RFCSA), and rectus femoris thickness (RF-Y-axis) in predicting malnutrition and sarcopenia in patients with esophagogastric cancer (EGC). Methods: This was a cross-sectional study of patients diagnosed with EGC. PhA was obtained using bioelectrical impedance vector analysis (BIVA) along with ASMMI. The RFCSA and RF-Y-axis were measured using nutritional ultrasound (NU^®^). Muscle capacity was assessed using handgrip strength (HGS), and functionality by applying the Short Physical Performance Battery (SPPB). Malnutrition and sarcopenia were determined according to the GLIM and EWGSOP2 criteria, respectively. Results: Out of the 35 patients evaluated, 82.8% had malnutrition and 51.4% had sarcopenia. The RFCSA (r = 0.582) and RF-Y-axis (r = 0.602) showed significant, moderate correlations with ASMMI, unlike PhA (r = 0.439), which displayed a weak correlation with this parameter. However, PhA (OR = 0.167, CI 95%: 0.047–0.591, *p* = 0.006), RFCSA (OR = 0.212, CI 95%: 0.074–0.605, *p* = 0.004), and RF-Y-axis (OR = 0.002, CI 95%: 0.000–0.143, *p* = 0.004) all showed good predicting ability for sarcopenia in the crude models, but only the RF-Y-axis was able to explain malnutrition in the regression model (OR = 0.002, CI 95%: 0.000–0.418, *p* = 0.023). Conclusions: The RF-Y-axis emerged as the only independent predictor of both malnutrition and sarcopenia in this study, likely due to its stronger correlation with ASMMI compared to PhA and RFCSA.

## 1. Introduction

According to the latest data from the Global Cancer Observatory (GLOBOCAN) of 2022, gastrointestinal (GI) cancers are a major public health concern, as they pose the highest lifetime risk of death due to the invasive nature of the disease [1]. In Europe, esophageal and gastric cancers, two of the most lethal malignant GI tumors [2,3], accounted for 189,031 new cases and 142,508 deaths in 2020 [4], resulting in a sixth and third place in terms of mortality, respectively [1].

These patients commonly experience a high rate of nutritional impairment due to symptoms arising from systemic inflammation and local tumor effects, such as dysphagia, nausea, malabsorption, vomiting, diarrhea, or fatigue [5,6,7]. This leads to inadequate nutritional intake [8,9], which causes involuntary weight loss and reduced muscle mass [10,11,12,13]. Therefore, disease-related malnutrition (DRM) and sarcopenia are the most common cancer-related conditions, affecting between 15% and 40% of patients at the time of diagnosis. Moreover, in advanced stages of esophagogastric cancer (EGC), DRM, and sarcopenia may affect up to 75% of patients [14,15,16,17].

Currently, in cancer patients, DRM and sarcopenia are associated with adverse outcomes, including a higher likelihood of postoperative complications and reduced response and tolerance to treatment [18]. This results in an increased length of hospital stay, disease burden, and healthcare costs, further worsening patient prognosis and overall survival [19,20,21]. Hypercatabolic states and, consequently, muscle wasting are often exacerbated by most chemotherapeutic agents and surgery itself, underscoring the importance of evaluating muscle mass as a key component in morphofunctional assessment to identify malnutrition and sarcopenia [22], which can also occur in individuals with normal weight, overweight, or obesity.

The Global Leadership Initiative on Malnutrition (GLIM) has highlighted the role of reduced muscle mass as a phenotypic criterion for diagnosing malnutrition in clinical settings [23]. Similarly, in 2019, the European Working Group on Sarcopenia in Older People 2 (EWGSOP2) updated the definition of this condition, establishing that sarcopenia is probable when low muscle strength is detected; its diagnosis is confirmed by the presence of low muscle quantity or quality and is considered severe when low physical performance is identified [24]. Several techniques are available to assess changes in body composition, such as bioelectrical impedance analysis (BIA) and ultrasound (US), which have the advantages of low cost, high portability, and bedside use [25,26], unlike magnetic resonance imaging (MRI), dual-energy X-ray absorptiometry (DEXA), and computerized tomography scan (CT), currently considered as gold standards for assessing the nutritional status of patients [27].

On the one hand, BIA is a non-invasive method based on the human body’s ability to transmit an electrical current, providing bioelectrical impedance vector analysis (BIVA) and phase angle (PhA), both of which elucidate insights into cell membrane integrity and vitality, and body hydration [28,29,30]. The BIVA approach and PhA are derived from raw measurements, specifically resistance (R) and reactance (Xc), rendering them independent of body weight and free from calculation-inherent errors, which makes them suitable for use in cancer patients [31].

On the other hand, although there is evidence that ultrasound muscle measurements are affected by fluid hydration status [32,33], US has recently proven to be a valuable tool for estimating muscle quantity and quality [34,35]. Although various muscular structures can be evaluated, the rectus femoris (RF) is one of the most referenced ones, since anterior thigh muscles are affected early in catabolic processes [36]. Like PhA, ultrasound-derived rectus femoris cross-sectional area (RFCSA) and muscle thickness or rectus femoris Y-axis (RF-Y-axis) have been proposed as attractive, effort-independent surrogate markers of malnutrition and sarcopenia. Recent studies have demonstrated that lower values of these parameters are linked to reduced muscle mass, strength and/or functionality [37,38,39,40].

However, contradictory findings have also been reported [41,42,43]. In addition, most studies yielding positive results have focused on contexts outside of oncology, including cardiovascular diseases, chronic obstructive pulmonary disease, SARS-CoV-2 disease, or even healthy patients [39,40,44]. Furthermore, research evaluating the effectiveness of PhA has predominantly examined its association with postoperative complication rates, length of hospital stay, quality of life, and survival, rather than with malnutrition and sarcopenia [45,46]. The same has occurred when considering US measurements, as studies have concentrated on establishing correlations between RFCSA, RF-Y-axis, and mortality [38].

Consequently, to date, there is a gap in the literature regarding the ability of certain bioelectrical and ultrasound parameters to reflect nutritional status, including muscle quantity and quality, in EGC patients. Therefore, under the hypothesis that PhA, RFCSA, and RF-Y-axis could play an important role in nutritional screening and subsequent diagnosis by detecting muscle loss, this study was aimed at evaluating the potential utility of PhA, RF-CSA, and RF-Y-axis in identifying malnutrition and sarcopenia according to the GLIM and EWGSOP2 criteria, respectively, in adult patients with EGC.

## 2. Materials and Methods

### 2.1. Study Design

This is a cross-sectional pilot study conducted as part of a prospective, single-center research project involving 35 patients diagnosed with esophagogastric cancer. Patients were recruited from the Endocrinology and Nutrition Service at the Hospital Universitario y Politécnico La Fe in Valencia between January and September 2024, after being referred from the departments of esophagogastric surgery or medical oncology.

The inclusion criteria were patients between 18 and 80 years old with histologically confirmed diagnoses, regardless of tumor stage or route of feeding. The exclusion criteria were patients with concomitant non-esophagogastric malignant tumors and those undergoing palliative treatment and patients with ECOG (Eastern Cooperative Oncology Group) >2. Patients diagnosed with severe liver cirrhosis, stage 4 or 5 chronic kidney disease (glomerular filtration rate less than 30 mL/min/1.73 m^2^, measured by the equation proposed by the Chronic Kidney Disease Epidemiology Collaboration (CKD-EPI)), heart failure, mental illness, or stroke were excluded. Additionally, patients without all clinical data, such as weight or height and contraindications to BIA, were also excluded. This ensured that, in total, 4 patients were excluded from the present study.

The study was approved by the Clinical Research Ethics Committee of the La Fe Health Research Institute (approval number 2023-1188-1, date of approval: 20 December 2023). Informed consent was obtained from all participants for the anonymous use of their data.

### 2.2. Clinical and Sociodemographic Data

We collected data related to sex, age, comorbidities such as diabetes, hypertension, dyslipemia, tumor location, tumor–node–metastasis (TNM) cancer staging system, oncology treatment, and ECOG. Information was recruited by interview or medical record. The level of physical activity was assessed using the International Physical Activity Questionnaire (IPAQ) [47]. Based on the results, the participants were classified into three groups: inactive or low physical activity, moderate activity, and high activity.

### 2.3. Anthropometric Measurements

Height was measured using a stadiometer, and weight was assessed with a calibrated weighing scale (SECA^®^, Hamburg, Germany), equipped with certified test weights (±0.1 kg). As part of anthropometry, the patients were measured with the patient standing, dressed in light clothing, barefoot, and with the head oriented in the Frankfurt horizontal plane, using a mechanical column scale. The body mass index (BMI) was calculated for each patient and classified according to the World Health Organization (WHO) guidelines. For older patients, the BMI was classified according to the recommendations of the Spanish Society of Gerontology and Geriatrics (SEGG) and the Spanish Society of Clinical Nutrition and Metabolism (SENPE) [48].

Calf (CC) and mid-arm circumferences (MAC) were measured according to recommendations using a flexible, non-elastic measuring tape (SECA*^®^* 201, Hamburg, Germany) calibrated in centimeters, with millimeter precision. The CC cut-off was set at <34 cm for men and <33 cm for women (BMI = 18.5–24.9 kg/m^2^), with adjustment factors applied for other BMI categories) [49].

### 2.4. Nutritional Screening and Diagnosis of Malnutrition

Nutritional risk was evaluated using subjective global assessment (SGA) [50]. SGA is the most studied, validated, and widely recognized method for accurately assessing the nutritional status of oncology patients [51,52]. It produces the following global ratings: well nourished (A), moderately malnourished (B), or severely malnourished (C).

The GLIM criteria were used to diagnose malnutrition [23], which requires the presence of at least one etiological and one phenotypic criterion simultaneously. The phenotypic criteria included: (a) unintentional weight loss > 5% over the past six months or >10% over a longer period, (b) a body mass index (BMI) < 18.5 kg/m^2^ for individuals under 70 years of age, or <20 kg/m^2^ for those aged 70 and older, and (c) reduced muscle mass based on appendicular skeletal muscle mass index (ASMMI) (<7 kg/m^2^ in males and <5.5 kg/m^2^ in females) or fat-free mass index (FFMI) (<17 kg/m^2^ in males and <15 kg/m^2^ in females).

We determined that all patients fulfilled the GLIM etiological criteria for chronic disease-related cancer. Dietary intake was estimated using a 24-h dietary recall of 3 days conducted by a trained registered dietician.

### 2.5. Morphofunctional Assessment

#### 2.5.1. Bioelectrical Impedance Vector Analysis (BIVA)

Impedance measurements were performed using a single-frequency, phase-sensitive impedance analyzer (NUTRILAB^®^, AKERN^®^, Pontassieve, Italy), which applies an alternating sinusoidal current of 400 µA at 50 kHz. The measurements were carried out following a standardized and validated technique [53] based on electrode placement (BIATRODES™, Pontassieve, Italy) on the back of the right hand (center of the third proximal phalanx) and on the corresponding foot (proximal to the second and third metatarsophalangeal joints). The position of the patients was supine, with the legs opened at a 45° angle relative to the body’s midline, while the upper limbs were positioned 30° away from the trunk. To avoid disturbances, all patients waited five minutes in a supine position to balance the fluid distribution, and they were instructed to abstain from food and drink for a 2 h period before the test [54]. Bioelectrical parameters were analyzed to estimate body composition, including PhA, total body water (TBW), extracellular body water (ECW), intracellular body water (ICW), fat-free mass (FFM), fat mass (FM), body cell mass (BMC), and appendicular skeletal muscle mass (ASMM). To assess the hydration status, the ECW/TBW ratio and TBW/FFM % were used.

#### 2.5.2. Nutritional Ultrasound (NU)^®^

The U PROBE-L6C^®^ (manufacturer Léleman^®^, Valencia, Spain) (linear 7.5–10 KHz) ultrasound scanner was used, as implemented by De Luis Román et al. in their disease-related caloric-protein malnutrition echography (DRECO) study [55]. The patient was in a relaxed supine position, with the knee fully extended. Ultrasound scans of the rectus femoris muscle were performed at a point two-thirds of the way between the superior pole of the patella and the anterior superior iliac spine, according to a standardized technique [56]. The probe was covered with a suitable water-soluble transmission gel to ensure proper acoustic contact without compressing the dermal surface. It was aligned perpendicularly to both the longitudinal and transverse axes of the rectus femoris muscle to acquire the transverse image (Figure 1A). We measured, in the transversal axis, the cross-sectional area (RFCSA) in cm^2^, muscle thickness (or RF-Y-axis), the RF-X-axis, and leg subcutaneous fat (RF-AT) in cm.

Adipose tissue assessment at the level of the abdominal wall was performed at the midpoint between the xiphoid process and the umbilicus. Cross-sectional imaging revealed the epidermis, superficial and deep adipose tissue layers, rectus abdominis muscles, and the preperitoneal fat layer between the *linea alba* and the parietal peritoneum. Measurements of total subcutaneous abdominal adipose tissue (T-SAT), superficial subcutaneous abdominal adipose tissue (S-SAT), and preperitoneal or visceral fat (VAT) were taken in centimeters during unforced expiration, perpendicularly to the skin. The procedure was carried out by a single experienced professional to minimize interobserver variability.

#### 2.5.3. Functional and Muscle Strength Assessment

Hand grip strength (HGS) in the dominant hand was measured using a Jamar dynamometer (J A Preston Corporation, New York, NY, USA). The patients were instructed to sit in a chair with a backrest, with both feet on the floor, with the shoulders close to the body in a neutral position and the forearm flexed at 90° without rotation [57,58]. The correct grip was then explained to them and initiated when they were in a comfortable position. They were asked to squeeze as hard as they could after receiving a verbal command; they were then verbally encouraged to achieve better results. Three measurements were recorded with the dominant hand, with 1 min of rest between each measurement, and then averaged.

Physical performance was assessed using the Short Physical Performance Battery (SPPB), which comprises three tests: balance (feet together, semi-tandem, and tandem), walking speed (over a 4-m distance), and the chair rise test. Based on the results, the patients were categorized as dependent/disabled (0–3 points), frail (4–6 points), pre-frail (7–9 points), or autonomous/robust (10–12 points) [59].

### 2.6. Assessment and Diagnosis of Sarcopenia

Sarcopenia risk was assessed using the validated Spanish version of the SARC-F [60], a five-item self-report questionnaire evaluating patients’ perceptions of their limitations in strength, assistance with walking, rising from a chair, climbing stairs, and experiences with falls. The final score was used to classify the patients as having a low probability of sarcopenia (<4 points) or a high probability of sarcopenia (≥4 points) [61].

Sarcopenia was diagnosed using the European Working Group on Sarcopenia in Older People (EWGSOP2) criteria [24]. Patients were classified according to the EWGSOP2 algorithm: (1) probable sarcopenia, defined by low muscle strength as measured by dynamometry (<27 kg in men and <16 kg in women); (2) confirmed sarcopenia, when low muscle strength coexists with low muscle quantity or quality, as determined by ASMMI (<7 kg/m^2^ in males and <5.5 kg/m^2^ in females); and (3) severe sarcopenia, when low strength and low muscle quantity/quality are accompanied by low physical performance (SPPB test ≤ 8 points).

### 2.7. Statistical Analysis

Continuous variables are presented as mean ± standard deviation (SD) or median with interquartile range (IQR), as appropriate. Categorical variables are expressed as proportions (%). Previously, the Shapiro–Wilk test was performed to check the normality of the data. Comparisons between groups were made with different tests, depending on the nature of the variables, including the Mann–Whitney U test, Fisher’s exact test, one-way ANOVA, and the Kruskal–Wallis test, followed by the Bonferroni post hoc test, as appropriate. Inferential statistics were performed with bivariate correlations using the Pearson and Spearman correlation tests, according to normal distribution. To determine whether different variables could predict malnutrition and sarcopenia, binary logistic regression analysis was conducted using a crude model, with the presence or absence of malnutrition and sarcopenia as the dependent variables. Statistical significance was set at *p* < 0.05. All statistical analyses were performed using SPSS version 30.0 (SPSS Inc., Chicago, IL, USA).

## 3. Results

The study included 35 patients, 26 of whom were male (74.3%), with a mean age of 62.8 ± 8.8 years. A total of 25 (71.4%) patients had esophageal cancer and 14 (40%) were in stage III. The combination of surgery and chemotherapy (CTx) was the most commonly applied therapy (54.3%). Most patients were inactive or considered to have low physical activity (74.3%). The characteristics of the population study, including demographic and clinical variables, screening methods, and anthropometric measurements, are summarized in Table 1.

Classical and advanced parameters of nutritional status assessment in the study sample, stratified by sex, are shown in Table 2.

According to SGA, 3 (8.6%) patients were classified as well nourished, 14 (40.0%) as mild to moderately malnourished, and 18 (51.4%) as severely malnourished. Similarly, 11 (31.4%) participants exhibited moderate malnutrition, and 18 (51.4%) presented severe malnutrition when applying the GLIM criteria, resulting in an overall malnutrition prevalence of 82.8%. Following the EWGSOP2 criteria, sarcopenia was observed in 18 patients (51.4%), with 8 (22.8%) classified as having confirmed sarcopenia and 10 (28.6%) as having severe sarcopenia, despite only 8 (22.8%) participants being identified as at risk for this condition based on SARC-F findings.

As shown in Table 3, weight loss was the only variable that showed a statistically significant difference between non-malnutrition and stages 1 (*p* = 0.010) and 2 (*p* < 0.001) of this condition. When analyzing BIVA-derived parameters by group pair, the PhA values associated with severe malnutrition (4.3 ± 0.7) were significantly lower than those corresponding to non-malnourished individuals (5.3 ± 0.7; *p* = 0.001). The data for BMI (*p* < 0.001), ASMMI (*p* = 0.003), and BCM (*p* = 0.017) exhibited the same trend.

Regarding US measurements, both RF-CSA (*p* < 0.001) and RF-Y-axis (*p* < 0.001) showed significant differences between the non-malnutrition and severe malnutrition groups. Although the %FM measured by BIVA did not reveal noteworthy variations among any of the groups, significant differences were observed between the RF adipose tissue values of the two groups (*p* = 0.020). Differences between malnutrition groups and sociodemographic and clinical variables such as physical activity, primary site tumor, comorbidities, tumor stage, and treatment were not found.

In relation to sarcopenia diagnosis, significant differences were observed in sex (*p* = 0.003) and age (*p* = 0.021) across the four groups, as well as in the SARC-F score. All patients without sarcopenia were men. Table 4 shows that the PhA values for individuals with confirmed (4.5 ± 0.8; *p* = 0.009) and severe sarcopenia (4.1 ± 0.5, *p* < 0.001) were significantly lower than those for patients without this condition (5.6 ± 0.7). This finding was also evident in the values obtained for BCM (*p* = 0.011, *p* < 0.001) and ASMMI (*p* < 0.001, *p* < 0.001). Like BCM and ASMMI, both RF-CSA and RF-y-axis showed significant differences between the non-sarcopenia group and the confirmed (*p* < 0.001) and severe sarcopenia groups (*p* = 0.027, *p* = 0.039).

Significant differences were also observed between the non-sarcopenia and sarcopenia probable groups when comparing the data for PhA (*p* = 0.044). The same pattern was noted for the other two diagnostic components of sarcopenia, showing significantly higher HGS values in patients without sarcopenia compared to those with probable (*p* = 0.037), confirmed (*p* = 0.002), and severe sarcopenia (*p* < 0.001). SPPB values in patients with severe sarcopenia were also significantly lower than those in the other groups (*p* < 0.001).

As with malnutrition, we did not find differences between sarcopenia groups and other sociodemographic and clinical variables.

All the primary variables examined—PhA, RFCSA, and RF-Y-axis—were significantly correlated with ASMMI, the key parameter used to diagnose malnutrition and sarcopenia (Figure 2). 

As shown in Table 5, the RF-Y-axis was the only muscle mass-related measure significantly correlated with all three diagnostic components of malnutrition, namely weight loss (r = −0.386, *p* = 0.022), BMI (r = 0.599, *p* < 0.001), and ASMMI (r = 0.602, *p* < 0.001). PhA was not correlated with either BMI or weight loss. In contrast, adipose tissue markers such as RF-AT (r = 0.742, *p* < 0.001), T-SAT (r = 0.826, *p* < 0.001), and S-SAT (r = 0.799, *p* < 0.001) showed high correlations with BMI, as well as FM (r = 0.543, *p* < 0.001), but low correlations with weight loss. These findings suggest that the RF-Y-axis may perform better than the other parameters as a predictor of malnutrition.

As detailed in Table 6, regarding the diagnostic components of sarcopenia, PhA exhibited statistically significant direct correlations with HGS (r = 0.556, *p* < 0.001), ASMMI (r = 0.439, *p* = 0.008), and SPPB (r = 0.475, *p* = 0.004), similar to BCM; however, the correlations with PhA were weaker. The RF-Y-axis showed the strongest association with ASMMI (r = 0.602, *p* < 0.001), but did not correlate with HGS, unlike RFCSA (r = 0.447, *p* = 0.007). Neither ultrasound measure was correlated with SPPB. These results suggest that PhA may provide a better prediction of sarcopenia than either the RFCSA or RF-Y-axis.

As shown in Figure 3, a positive correlation was found between RF ultrasound measurements and BIVA-derived parameters. RFCSA showed a moderate positive correlation with PhA (r = 0.564, *p* < 0.001) and BCM (r = 0.533, *p* < 0.001). The RF-Y-axis revealed a weak positive correlation with PhA (r = 0.457, *p* = 0.006) and BCM (r = 0.445, *p* = 0.007).

The unadjusted binary logistic regression models aimed at predicting the presence of malnutrition demonstrated that higher values of the RF-Y-axis (OR = 0.002, IC 95%: 0.000–0.418, *p* = 0.023) are protective factors against this condition. As shown in Table 7, for each one-cm increase in the RF-Y-axis, the likelihood of not having malnutrition is 500 times higher. PhA, RFCSA, and RF-AT failed to predict malnutrition in this case.

Then, the crude analyses for predicting sarcopenia indicated that higher values of PhA (OR = 0.167, IC 95%: 0.047–0.591, *p* = 0.006) and ultrasound measurements of the rectus femoris, namely RFCSA (OR = 0.212, IC 95%: 0.074–0.605, *p* = 0.004) and RF-Y-axis (OR = 0.002, IC 95%: 0.000–0.143, *p* = 0.004), are protective against this condition. Specifically, the likelihood of being free from sarcopenia increases by 5.99 times with each one-degree increase in PhA. Similarly, for every one-centimeter increase in RFCSA and the RF-Y-axis, the probability of not having sarcopenia rises by 4.72 and 500 times, respectively.

It is worth noting that BCM, like the RF-Y-axis, showed good predictive ability in both crude models for malnutrition and sarcopenia. However, the estimation of this parameter relies on predictive BIA equations, which require data such as weight and height that are not always available. For this reason, attention has been focused on the results corresponding to PhA, RFCSA, and RF-Y-axis. Multivariable logistic regression models were not conducted due to the limited sample size.

## 4. Discussion

To the best of our knowledge, this is the first study to investigate the potential usefulness of phase angle and nutritional ultrasound in identifying the presence of malnutrition and sarcopenia in European patients with EGC using the most recent diagnostic criteria (GLIM and EWGSOP2). Only two studies have assessed the predictive value of PhA in patients with gastrointestinal cancer, one focusing solely on malnutrition [62] and the other including sarcopenia [63]. In fact, when considering ultrasound, only one study used RF-CSA and RF-Y-axis to predict these two deleterious conditions in head and neck cancer patients [64], while another one used it to anticipate 12-month mortality in a similar sample [38].

Our investigation identified that malnutrition was highly prevalent in esophageal and gastric cancer patients (82.8%), with 31.4% of patients showing moderate malnutrition and 51.4% with severe malnutrition. These values are higher than those found in most studies with the same population and similar methodology [63,65,66,67]. Moreover, these investigations have emphasized that patients who are candidates for oncological surgery, such as most of those included in our study, are twice as likely to present with malnutrition. A study recorded 72.2% malnutrition in patients after esophagogastric cancer surgery [68].

Moreover, this research showed that sarcopenia was highly prevalent in the patients analyzed with EGC, representing 51.5% of them. As with undernutrition, these results are significantly higher compared to other studies [69,70]. The discrepancies observed can primarily be attributed to differences in methodology, as most studies have used different diagnostic criteria or another technique to assess body composition, such as CT scans. Only one study included the EWGSOP2 diagnostic algorithm for sarcopenia, which found 43.3% sarcopenic patients [71]. However, in studies that included patients who underwent esophagectomy or gastrectomy [72,73], the prevalence of sarcopenia increased considerably (57.4% and 57.7%, respectively), more closely resembling our results.

Clinical characteristics, such as tumor site, tumor stage, and type of treatment, did not show significant differences between the malnutrition and sarcopenia groups, likely due to sample heterogeneity, which resulted in very small frequencies in each subgroup. However, statistically significant differences were observed in some BIVA-derived parameters, such as PhA, ASMMI, and BCM. This trend has also been recorded in multiple studies carried out in oncology patients [29,74,75]. PhA was positively correlated with all the components of sarcopenia diagnosis (ASMMI, HGS, and SPPB). Zuo et al. previously reported a similar correlation in gastric cancer patients [63]. Unlike the results observed in our study, they also found a positive correlation between PhA and all the nutritional indices used to diagnose malnutrition according to the GLIM criteria.

Interestingly, BCM was the parameter most strongly correlated with the diagnostic components of malnutrition and sarcopenia. Also, the crude analyses for predicting these two conditions demonstrated that a higher value of BCM is a protective factor against malnutrition and sarcopenia. These results are consistent with those reported by Herrera-Martínez et al. in a large cohort of patients with head and neck cancer [76]. Their results demonstrated that BCM was more strongly associated with malnutrition (OR = 0.88, 95% CI = 0.84–0.93, *p* < 0.001) and sarcopenia (OR = 0.81, 95% CI = 0.76–0.87, *p* < 0.001) compared to PhA (OR = 0.54, 95% CI = 0.40–0.71, *p* < 0.001) (OR = 0.47, 95% CI = 0.33–0.66, *p* < 0.001).

However, the present study focused on parameters such as PhA given its clinical significance, but in our study, phase angle was not able to predict malnutrition, although it could predict sarcopenia. Conversely, the study by Yang et al., using logistic regression models, confirmed PhA as a valuable indicator of malnutrition in patients with gastrointestinal cancer (OR = 0.548, 95% CI = 0.385–0.780, *p* < 0.001) [62]. A potential explanation for the discrepancies could be the altered hydration status and the small size of our study sample. The mean ECW/TBW index that we found exceeded the reference value established by Ge et al. [77] for the oncologic population with sarcopenia, evidencing a state of overhydration (ECW/TBW ≥ 0.385), which may interfere with correlations involving PhA.

The use of NU^®^-derived parameters based on muscle area and thickness (RFCSA and RF-Y-axis, respectively) may contribute to the assessment of malnutrition and sarcopenia. We found a moderate positive correlation between RFCSA and R-Y-axis with ASMMI, as previously described by Hida et al. [78]. Like Lopez-Gómez et al. [79], we also detected a weak correlation between RFCSA and HGS, which indicates that RF ultrasound measurements could be related not only to muscle quantity but also to muscle strength. This is supported by previous research on the role of ultrasound in the prediction of sarcopenia in elderly patients. It was revealed that the RFCSA and RF-Y-axis were the best indicators for detecting the loss of muscle mass and strength [80].

The RF-Y-axis was the only marker capable of predicting both sarcopenia and malnutrition. Furthermore, it exhibited the strongest correlation with ASMMI when considering PhA and RFCSA. Ozturk et al. also disclosed that the RF-Y-axis had a slightly greater positive correlation with skeletal muscle mass for the diagnosis of malnutrition using GLIM criteria in hospitalized internal medicine patients [81].

Due to the limited literature using these ultrasound measurements as markers of malnutrition and sarcopenia, making direct comparisons was challenging. In cancer patients, we have only the data reported by two Spanish studies [64,79]. On the one hand, Fernández-Jiménez et al. described that high levels of the RFCSA (OR = 0.81 (0.68–0.98), *p* < 0.05) and RF-Y-axis (OR = 0.31 (0.15–0.61), *p* < 0.001) were associated with a decreased risk of malnutrition, as defined by the GLIM criteria. Sarcopenia showed the same trend (OR = 0.64 (0.49–0.82), *p* < 0.001) for RFCSA and (OR = 0.27 (0.11–0.68), *p* < 0.01) for RF-Y-axis. On the other hand, Lopez-Gómez only found statistical differences in the RFCSA with sarcopenia diagnosis (sarcopenia: 2.47 cm^2^ (±0.54 cm^2^); no sarcopenia: 3.65 cm^2^ (±1.34 cm^2^); *p* = 0.02), but no differences with malnutrition.

Concerning adipose tissue, assessed using NU^®^, we found that all abdominal measurements (T-SAT, S-SAT, and VAT) and RF adipose tissue were significantly different between malnutrition groups. Additionally, T-SAT and S-SAT were correlated with all the components of malnutrition diagnosis, and they could predict malnutrition in the crude logistic regression analysis. As expected and described by other studies [64], US adipose tissue measurements did not show any relation with sarcopenia parameters, since they are highly associated with methods of assessing fat deposition and distribution.

Furthermore, we found a significant correlation between RFCSA and R-Y-axis with PhA, BCM, and ASMMI, which is consistent with a previous study in a longitudinal cohort of patients with cancer [38] and with the DRECO study [55]. These findings may support the integration of BIVA and NU^®^ as part of the morphofunctional assessment for monitoring and optimizing the nutritional status of cancer patients, with both techniques being easily accessible in routine clinical practice.

There were several limitations to the current study. First, this was a cross-sectional study with a small sample size and a low proportion of women. Multicenter trials including a larger number of patients with EGC are needed for further validation. Second, multivariate logistic regression models were not conducted, which weakened the results by excluding important confounding factors such as age, type of treatment, and BMI. Third, the lack of consensus regarding cut-off values for PhA, RFCSA, RF-Y-axis, and RF-AT limited the ability to compare the results with previous studies. In addition, the cross-sectional nature of the study prevented participant follow-up. Therefore, prospective studies that include patients with different types of GI cancer are essential in order to establish causal relationships derived from nutritional intervention, thereby obtaining results that can be extrapolated to the oncological population.

## 5. Conclusions

In conclusion, the RF-Y-axis is the only parameter that appears to be a promising and useful independent predictor of both malnutrition and sarcopenia in this sample of EGC patients. These results reinforce the implementation of RF-Y-axis in routine clinical practice and its use as a potential low muscle quantity or quality criterion in the EWGSOP2 criteria and as a potential phenotypic criterion for muscle mass loss in the GLIM criteria. Nevertheless, PhA and RFCSA demonstrated good performance in predicting sarcopenia, but not malnutrition in the same population. This suggests the need for a larger sample to demonstrate stronger correlations between these two markers and ASMMI in order to effectively determine their usefulness as predictors not only of the presence but also of the severity of malnutrition and sarcopenia.

This study represents the initial exploration of an ongoing prospective nutritional follow-up project aimed at improving the process of identifying patients who require multimodal interventions, as well as assessing the outcomes of these interventions in terms of body composition and function. In this way, the research conducted would allow the results obtained to be translated into a more practical, effective, and objective morphofunctional assessment, thereby supporting the work of health professionals.

## Figures and Tables

**Figure 1 nutrients-17-00091-f001:**
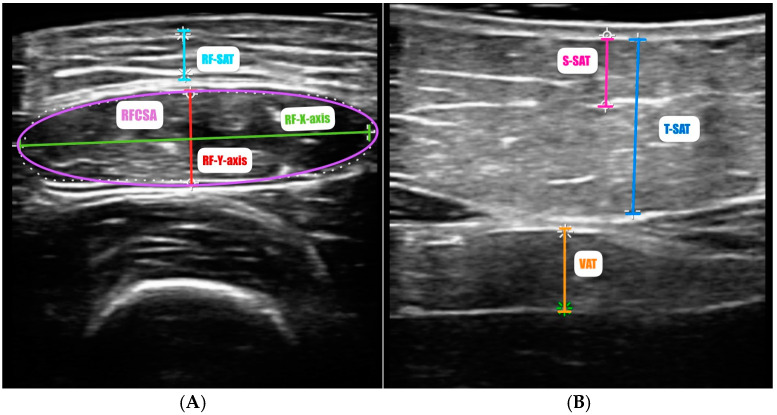
Measurement of rectus femoris (**A**) and abdominal adipose tissue (**B**) by ultrasound of a patient in our sample. Abbreviations—RFCSA: rectus femoris cross-sectional area; RF-Y-axis: rectus femoris Y-axis or muscle thickness; RF-X-axis: rectus femoris X-axis; RF-SAT: rectus femoris superficial adipose tissue; VAT: visceral adipose tissue; T-SAT: total subcutaneous adipose tissue; S-SAT: superficial subcutaneous adipose tissue.

**Figure 2 nutrients-17-00091-f002:**
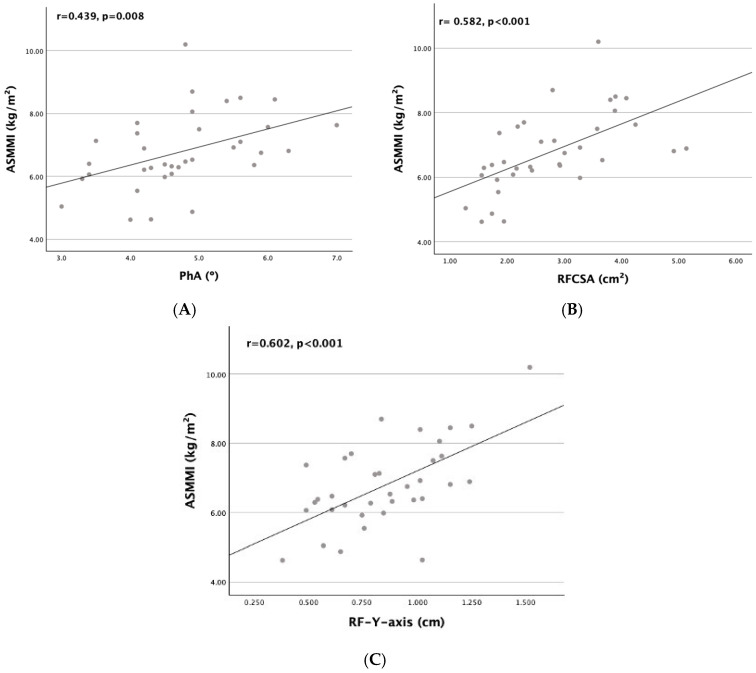
Scatter plot graphs of correlation between ASMMI and (**A**) PhA, (**B**) RFCSA, and (**C**) RF-Y-axis.

**Figure 3 nutrients-17-00091-f003:**
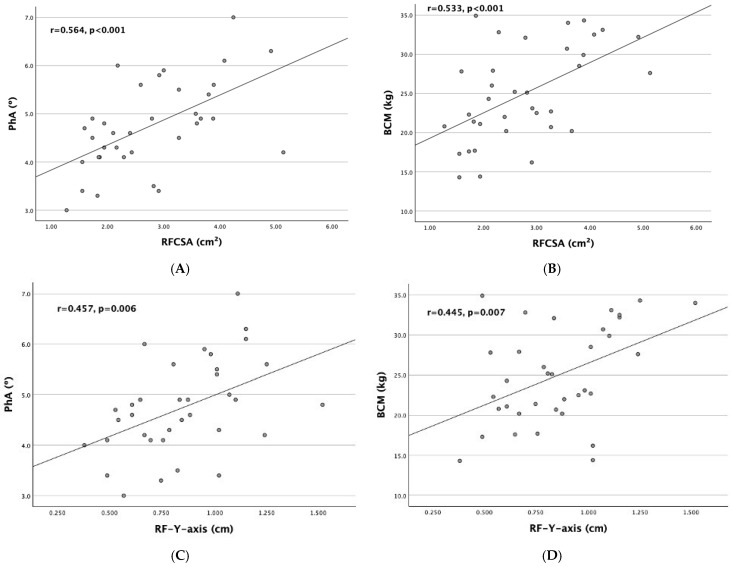
Scatter plot graphs of correlation between RF ultrasound measurements and BIVA-derived parameters: (**A**) RFCSA with PhA, (**B**) RFCSA with BCM, (**C**) RF-Y-axis with PhA, and (**D**) RF-Y-axis with BCM.

**Table 1 nutrients-17-00091-t001:** Baseline demographic and disease characteristics of the participants.

Variable	All Patients (*n* = 35)	Male (*n* = 26)	Female (*n* = 9)
**Age (years)**	62.8 ± 8.8	62.2 ± 9.5	64.8 ± 6.4
**Primary site tumor**			
Esophageal	25 (71.4%)	21 (80.8%)	4 (44.4%)
Gastric	10 (28.6%)	5 (19.2%)	5 (55.6%)
**Tumor stage**			
I	3 (8.6%)	0 (0%)	3 (33.3%)
II	10 (28.6)	8 (30.8%)	2 (22.2%)
III	14 (40%)	11 (42.3%)	3 (33.3%)
IV	8 (22.9%)	7 (26.9%)	1 (11.1%)
**Comorbidities**			
0	7 (20%)	6 (23.1%)	1 (11.1%)
1	8 (22.9%)	6 (23.1%)	2 (22.2%)
≥2	20 (57.1%)	14 (53.8%)	6 (66.7%)
**Physical activity**			
Low or inactive	26 (74.3%)	19 (73.1%)	7 (77.8%)
Medium	5 (14.3%)	3 (11.5%)	2 (22.2%)
High	4 (11.4%)	4 (15.4%)	0 (0%)
**Treatment**			
Only CTx	8 (22.9%)	8 (30.8%)	0 (0%)
CTx and RTx	4 (11.4%)	2 (7.7%)	2 (22.2%)
Surgery and CTx	10 (54.3%)	13 (50%)	6 (66.7%)
Surgery, CTx and RTx	4 (11.4%)	3 (11.5%)	1 (11.1%)

Data are expressed as mean ± standard deviations or percentage. The groups were divided by sex variable. Abbreviations—CTx: chemotherapy; RTx: radiotherapy.

**Table 2 nutrients-17-00091-t002:** Morphofunctional assessment parameters stratified by sex.

Variable	All Patients (*n* = 35)	Male (*n* = 26)	Female (*n* = 9)
**BMI (kg/m^2^)**	23.3 ± 5.7	23.5 ± 5.3	22.6 ± 6.9
Underweight	13 (37.1%)	9 (34.6%)	4 (44.4%)
Normal	12 (34.3%)	10 (38.5%)	2 (22.2%)
Overweight	4 (11.4%)	3 (11.5%)	1 (11.1%)
Obesity	6 (17.1%)	4 (15.4%)	2 (22.2%)
**Weight loss within past 6 months (%)**	14.3 ± 7.9	14.9 ± 8.3	12.3 ± 6.5
<5%	4 (11.4%)	3 (11.5%)	1 (11.1%)
5–10%	5 (14.3%)	2 (7.7%)	3 (33.3%)
>10%	26 (74.3%)	21 (80.8%)	5 (55.6%)
**MAC (cm)**	26.1 ± 5.3	23.5 ± 5.3	25.2 ± 6.9
**CC (cm)**	32.9 ± 4.4	33.3 ± 4.5	31.9 ± 4.1
Normal	8 (22.9%)	6 (23.1%)	2 (22.2%)
Low	27 (77.1%)	20 (66.9%)	7 (77.8%)
**BIVA-derived parameters**			
PhA (°)	4.7 ± 0.9	4.9 ± 0.9	4.3 ± 0.8
ECW/TBW ratio	0.5 ± 0.07	0.48 ± 0.06	0.51 ± 0.08
TBW/FFM (%)	69.7 ± 17.6	71.1 ± 14.7	66.0 ± 24.6
FM (%)	19.6 ± 12	18.7 ± 10.2	22.2 ± 16.6
ASMMI (kg/m^2^)	6.8 ± 1.2	7.2 ± 1.06	5.6 ± 0.86
BCM (kg)	24.9 ± 6.1	27.1 ± 5.2	18.8 ± 3.9
**Nutritional ultrasound^®^: rectus femoris muscle**			
RFCSA (cm^2^)	2.8 ± 1.0	2.9 ± 1.02	2.2 ± 0.8
RF-Y-axis (cm)	0.8 ± 0.3	0.87 ± 0.27	0.77 ± 0.22
RF-X-axis	3.65 ± 0.50	3.76 ± 0.44	3.31 ± 0.55
RF-AT (cm)	0.41 (0.23–0.74)	0.35 (0.24–0.55)	0.78 (0.22–1.42)
**Nutritional ultrasound^®^: abdominal adipose tissue**			
T-SAT (cm)	1.4 (0.5–1.9)	1.35 (0.47–1.85)	1.41 (0.82–2.43)
S-SAT (cm)	0.52 (0.28–0.87)	0.47 (0.26–0.79)	0.68 (0.35–1.06)
VAT (cm)	0.55 (0.31–0.73)	0.52 (0.30–0.65)	0.58 (0.33–0.95)
**Hand grip strength**			
HGS (kg)	27.5 ± 8.4	31.1 ± 6.5	17.3 ± 2.5
**Functional test**			
SPPB	10 (7–11)	10 (7.7–11.2)	10 (6.5–10.5)

Data are expressed as mean ± standard deviation or median (interquartile range). Abbreviations—BMI: body mass index; MAC: mid-arm circumference; CC: calf circumference;; BIVA: bioelectrical impedance vector analysis; PhA: phase angle; ECW: extracellular water; TBW: total body water; FFM: fat-free mass; FM: fat mass; ASMMI: appendicular skeletal muscle mass index; BCM: body cellular mass; RFCSA: rectus femoris cross-sectional area; RF-Y-axis: rectus femoris Y-axis; RF-X-axis: rectus femoris X-axis; RF-AT: rectus femoris adipose tissue; T-SAT: total subcutaneous adipose tissue; S-SAT: superficial subcutaneous adipose tissue; VAT: visceral adipose tissue; HGS: hand grip strength; SPPB: Short Physical Performance Battery.

**Table 3 nutrients-17-00091-t003:** Differences in demographic, clinical, BIVA-derived, and ultrasound data according to the GLIM criteria.

Variable	No Malnutrition (*n* = 6)	Moderate Malnutrition (*n* = 11)	Severe Malnutrition (*n* = 18)	*p*-Value
**Sex**				0.773
Male	4 (66.7%)	9 (81.8%)	13 (72.2%)	
Female	2 (33.3%)	2 (18.2%)	5 (27.8%)	
**Age (years)**	60.5 ± 4.8	63.3 ± 10.9	63.3 ± 8.7	0.786
**BMI (kg/m^2^)**	29.3 ± 5.6	25.9 ± 4.8	19.6 ± 3.0	<0.001 ***
**Weight loss within past 6 months (%)**	4.2 ± 4.3	14.9 ± 5.5	17.3 ± 7.4	<0.001 ***
**MAC (cm)**	31.0 ± 4.6	29.4 ± 4.3	22.4 ± 3.0	<0.001 ***
**CC (cm)**	36.8 ± 5.3	34.9 ± 3.5	30.3 ± 2.8	<0.001 ***
**SGA**				
Well nourished (A)	3 (50%)	0 (0%)	0 (0%)	<0.001 ***
Mild to moderately malnourished (B)	2 (33.3%	8 (72.7%)	4 (22.2%)	
Severely malnourished (C)	1 (16.7%)	3 (27.3%)	14 (77.8%)	
**BIVA-derived parameters**				
PhA (°)	5.3 ± 0.7	5.1 ± 1.02	4.3 ± 0.7	0.016 *
ECW/TBW ratio	0.46 ± 0.04	0.47 ± 0.07	0.50 ± 0.07	0.405
TBW/FFM (%)	62.3 ± 30.3	74.3 ± 2.1	69.7 ± 17.2	0.430
FM (%)	25.7 ± 13.9	21.0 ± 13.6	16.7 ± 9.8	0.255
ASMMI (kg/m^2^)	7.7 ± 1.5	7.6 ± 0.8	6.0 ± 0.8	<0.001 ***
BCM (kg)	29.9 ± 4.7	27.6 ± 5.6	21.7 ± 4.9	<0.001 ***
**Nutritional ultrasound^®^: rectus femoris muscle**				
RFCSA (cm^2^)	3.5 ± 0.9	3.5 ± 0.9	2.1 ± 0.6	<0.001 ***
RF-Y-axis (cm)	1.1 ± 0.3	0.97 ± 0.19	0.68 ± 0.18	<0.001 ***
RF-X-axis (cm)	3.64 ± 0.22	3.96 ± 0.12	3.46 ± 0.11	0.030 *
RF-AT (cm)	0.82 (0.4–1.16)	0.44 (0.34–1.01)	0.30 (0.18–0.50)	0.037 *
**Nutritional ultrasound^®^:** **abdominal adipose tissue**				
T-SAT (cm)	2.08 (1.72–2.59)	1.30 (0.55–2.62)	1.0 (0.36–1.56)	0.012 *
S-SAT (cm)	1.07 (0.73–1.26)	0.62 (0.34–0.95)	0.44 (0.19–0.65)	0.011 *
VAT (cm)	0.63 (0.56–0.97)	0.62 (0.55–0.93)	0.34 (0.24–0.48)	0.004 **

Data are expressed as mean ± standard deviation or median (interquartile range) or percentage. Asterisk indicates significant difference between groups, according to the Mann–Whitney test or Fisher’s exact test (*** *p* < 0.001, ** *p* < 0.01, * *p* < 0.05). Abbreviations—BMI: body mass index; MAC: mid-arm circumference; CC: calf circumference; SGA: subjective global assessment; BIVA: bioelectrical impedance vector analysis; PhA: phase angle; ECW: extracellular water; TBW: total body water; FFM: fat-free mass; FM: fat mass; ASMMI: appendicular skeletal muscle mass index; BCM: body cellular mass; RFCSA: rectus femoris cross-sectional area; RF-Y-axis: rectus femoris Y-axis; RF-X-axis: rectus femoris X-axis; RF-AT: rectus femoris adipose tissue; T-SAT: total subcutaneous adipose tissue; S-SAT: superficial subcutaneous adipose tissue; VAT: visceral adipose tissue.

**Table 4 nutrients-17-00091-t004:** Differences in demographic, clinical, BIVA-derived, and ultrasound data according to the EWGSOP2 criteria.

Variable	No Sarcopenia (*n* = 12)	Probable Sarcopenia (*n* = 5)	Confirmed Sarcopenia (*n* = 8)	Severe Sarcopenia (*n* = 10)	*p*-Value
**Sex**					0.003 **
Male	12 (100%)	1 (20%)	5 (62.5%)	8 (80%)	
Female	0 (0%)	4 (80%)	3 (37.5%)	2 (20%)	
**Age (years)**	57.7 ± 9.4	68.8 ± 6.5	61.2 ± 6.9	67.2 ± 7.1	0.021 *
**BMI (kg/m^2^)**	27.3 ± 5.2	26.9 ± 5.5	18.5 ± 3.2	20.3 ± 2.6	<0.001 ***
**MAC (cm)**	30.4 ± 3.8	29.2 ± 5.5	20.4 ± 2.3	23.9 ± 2.4	<0.001 ***
**CC (cm)**	36.7 ± 4.2	33.3 ± 3.4	30.3 ± 3.1	30.3 ± 2.5	<0.001 ***
**SARC-F**					<0.001 ***
No risk	12 (100%)	2 (40%)	8 (100%)	5 (50%)	
Sarcopenia risk	0 (0%)	3 (60%)	0 (0%)	5 (50%)	
**BIA-derived parameters**					
PhA (°)	5.6 ± 0.7	4.5 ± 0.9	4.5 ± 0.8	4.1 ± 0.5	<0.001 ***
ECW/TBW ratio	0.47 ± 0.04	0.48 ± 0.09	0.48 ± 0.07	0.51 ± 0.07	0.550
TBW/FFM (%)	74.15 ± 1.97	60.13 ± 33.3	73.4 ± 0.29	66.0 ± 24.5	0.407
FM (%)	19.4 ± 12.7	25.2 ± 19.1	14.2 ± 8.9	21.4 ± 8.4	0.408
ASMMI (kg/m^2^)	7.98 ± 0.95	6.73 ± 0.57	5.92 ± 0.96	6.08 ± 0.69	<0.001 ***
BCM (kg)	30.2 ± 3.6	23.5 ± 6.3	22.7 ± 6.3	21.1 ± 3.9	<0.001 ***
**Nutritional ultrasound^®^: rectus femoris muscle**					
RFCSA (cm^2^)	3.56 ± 0.76	2.80 ± 0.60	1.82 ± 0.49	2.53 ± 1.05	<0.001 ***
RF-Y-axis (cm)	1.05 ± 0.22	0.86 ± 0.13	0.58 ± 0.18	0.80 ± 0.21	<0.001 ***
RF-X-axis (cm)	3.79 ± 0.11	3.75 ± 0.20	3.49 ± 0.99	3.55 ± 0.23	0.520
RF-AT (cm)	0.48 (0.36–0.85)	1.10 (0.28–1.84)	0.26 (0.15–0.48)	0.30 (0.18–0.49)	0.072
**Hand grip strength**					
HGS (kg)	35.9 ± 4.9	19.5 ± 7.1	25.4 ± 5.2	23.3 ± 6.2	<0.001 ***
**Functional test**					
SPPB	11.5 (10–12)	10 (7.5–10.5)	10 (9–10.75)	6 (5–7.25)	<0.001 ***

Data are expressed as mean ± standard deviation or median (interquartile range) or percentage. Asterisk indicates significant difference between groups, according to the Mann–Whitney test or Fisher’s exact test (*** *p* < 0.001, ** *p* < 0.01, * *p* < 0.05) Abbreviations—BMI: body mass index; MAC: mid-arm circumference; CC: calf circumference; BIVA: bioelectrical impedance vector analysis; PhA: phase angle; ECW: extracellular water; TBW: total body water; FFM: fat-free mass; FM: fat mass; ASMMI: appendicular skeletal muscle mass index; BCM: body cellular mass; RFCSA: rectus femoris cross-sectional area; RF-Y-axis: rectus femoris Y-axis; RF-X-axis: rectus femoris X-axis; RF-AT: rectus femoris adipose tissue; HGS: hand grip strength; SPPB: Short Physical Performance Battery.

**Table 5 nutrients-17-00091-t005:** Correlations between BIVA-derived parameters, ultrasound measurements, and components of malnutrition diagnosis according to GLIM criteria (%weight loss, BMI, and ASMMI).

Variable	% Weight Loss		BMI (kg/m^2^)		ASMMI (kg/m^2^)	
	r	*p*-Value	r	*p*-Value	r	*p*-Value
**BIA-derived parameters**						
PhA (°)	−0.093	0.596	0.288	0.094	0.439	0.008 **
ECW/TBW ratio	0.244	0.157	0.244	0.158	−0.246	0.154
TBW/FFM (%)	0.156	0.378	0.261	0.136	0.169	0.338
FM (%)	−0.232	0.181	0.543	<0.001 ***	0.204	0.240
BCM (kg)	−0.313	0.067	0.397	0.018 *	0.837	<0.001 ***
**Nutritional ultrasound^®^: rectus femoris muscle**						
RFCSA (cm^2^)	−0.311	0.069	0.531	0.001 ***	0.582	<0.001 ***
RF-Y-axis (cm)	−0.386	0.022 *	0.599	<0.001 ***	0.602	<0.001 ***
RF-AT (cm)	−0.420	0.012 *	0.742	<0.001 ***	0.245	0.156
**Nutritional ultrasound^®^: abdominal adipose tissue**						
T-SAT (cm)	−0.491	0.005 **	0.826	<0.001 ***	0.399	0.026 *
S-SAT (cm)	−0.459	0.009 **	0.799	<0.001 ***	0.416	0.020 *
VAT (cm)	−0.092	0.112	0.607	<0.001 ***	0.278	0.130

Asterisk indicates significant correlation (*** *p* < 0.001, ** *p* < 0.01, * *p* < 0.05). Abbreviations—BMI: body mass index; ASMMI: appendicular skeletal muscle mass index; PhA: phase angle; ECW: extracellular water; TBW: total body water; FFM: fat-free mass; FM: fat mass; BCM: body cellular mass; RFCSA: rectus femoris cross-sectional area; RF-Y-axis: rectus femoris Y-axis; RF-AT: rectus femoris adipose tissue; T-SAT: total subcutaneous adipose tissue; S-SAT: superficial subcutaneous adipose tissue; VAT: visceral adipose tissue.

**Table 6 nutrients-17-00091-t006:** Correlations between BIVA-derived parameters, ultrasound measurements, and components of sarcopenia diagnosis according to the EWGSOP2 (HGS, ASMMI, SPPB).

Variable	HGS (kg)		ASMMI (kg/m^2^)		SPPB	
	r	*p*-Value	r	*p*-Value	r	*p*-Value
**BIA-derived parameters**						
PhA (°)	0.556	<0.001 ***	0.439	0.008 **	0.475	0.004 **
ECW/TBW ratio	−0.257	0.135	−0.246	0.154	−0.376	0.026 *
TBW/FFM (%)	0.223	0.204	0.169	0.338	0.124	0.483
BCM (kg)	0.751	<0.001 ***	0.837	<0.001 ***	0.461	0.005 **
**Nutritional ultrasound^®^: rectus femoris muscle**						
RFCSA (cm^2^)	0.447	0.007 **	0.582	<0.001 ***	0.233	0.178
RF-Y-axis (cm)	0.315	0.065	0.602	<0.001 ***	0.151	0.388

Asterisk indicates significant correlation (*** *p* < 0.001, ** *p* < 0.01, * *p* < 0.05). Abbreviations—HGS: hand grip strength; ASMMI: appendicular skeletal muscle mass index; SPPB: Short Physical Performance Battery; PhA: phase angle; ECW: extracellular water; TBW: total body water; FFM: fat-free mass; BCM: body cellular mass; RFCSA: rectus femoris cross-sectional area; RF-Y-axis: rectus femoris Y-axis.

**Table 7 nutrients-17-00091-t007:** Crude logistic regression analysis evaluating PhA, RFCSA, RFT, T-SAT, and S-SAT with GLIM malnutrition and EWGSOP2 sarcopenia.

	Malnutrition		Sarcopenia	
Variables	OR	*p*-Value	OR	*p*-Value
PhA (°)	0.430 (0.152–1.217)	0.112	0.167 (0.047–0.591)	0.006 **
BCM (kg)	0.817 (0.673–0.993)	0.042 *	0.797 (0.682–0.932)	0.005 **
**Nutritional ultrasound^®^: rectus femoris muscle**				
RFCSA (cm^2^)	0.401 (0.155–1.037)	0.060	0.212 (0.074–0.605)	0.004 **
RF-Y-axis (cm)	0.002 (0.000–0.418)	0.023 *	0.002 (0.000–0.143)	0.004 **
RF-AT (cm)	0.220 (0.035–1.369)	0.105		
**Nutritional ultrasound^®^: abdominal adipose tissue**				
T-SAT (cm)	0.192 (0.043–0.851)	0.030 *		
S-SAT (cm)	0.019 (0.001–0.448)	0.014 *		

Asterisk indicates statical significance (** *p* < 0.01, * *p* < 0.05). Abbreviations—PhA: phase angle; BCM: body cellular mass; RFCSA: rectus femoris cross-sectional area; RF-Y-axis: rectus femoris Y-axis; RF-AT: rectus femoris adipose tissue; T-SAT: total subcutaneous adipose tissue; S-SAT: superficial subcutaneous adipose tissue.

## Data Availability

Data are contained within the article.

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
