# Peer review of "Phase Angle and Ultrasound Assessment of the Rectus Femoris for Predicting Malnutrition and Sarcopenia in Patients with Esophagogastric Cancer: A Cross-Sectional Pilot Study"

_nutrients, 2024, doi:10.3390/nu17010091_

Round 1
Reviewer 1 Report
Comments and Suggestions for Authors
This study aimed to evaluate the potential utility of PhA, RFCSA and RF-Y-25 axis in predicting malnutrition and sarcopenia in patients with esophagogastric cancer (EGC). This study is very great for readers and scientific community. However, several concerns are arised.
Abstract:
Background is too long.
Introduction:
US has significant impact on hydration, differently that authors described in the introduction.
What is hypothesis of study?
Whats are studies that used the The U PROBE-L6C® manufacturer Léleman®. Literature has approved this tool? in cancer patients?
Methods:
What is sample size calculus? This is a SMALL sample size!
These patients are using medicines?
There are food consumption data?
Results:
Several parts are results description are a repetition of tables.
Crude logistic with small sample size is speculative!
Why BMI is limitation? Dont make sense. Small sample size is a big limitation!
Author Response
Reviewer 1
This study aimed to evaluate the potential utility of PhA, RFCSA and RF-Y-25 axis in predicting malnutrition and sarcopenia in patients with esophagogastric cancer (EGC). This study is very great for readers and scientific community. However, several concerns are arised.
Abstract:
Background is too long.
Dear reviewer
Thank you very much for your comment. We have proceeded to reduce the background section to make it less extensive.
Introduction:
US has significant impact on hydration, differently that authors described in the introduction.
What is hypothesis of study?
We thank you once again for your enriching comments.
Regarding the hypothesis, we have revised the study hypothesis to make the study more easily understandable.
Whats are studies that used the The U PROBE-L6C® manufacturer Léleman®. Literature has approved this tool? in cancer patients?
Thank you for your insightful question. The U PROBE-L6C® device has been used in several studies assessing muscle mass and functional evaluation, also in cancer patients. InIn Spain, one of the largest studies validating the use of this type of ultrasound device is the DRECO (Disease-Related caloric-protein malnutrition EChOgraphy) study, a multicenter study that assessed hospitalized patients with disease-related malnutrition to establish cutoff points for RFCSA and RF-Y-axis measurements for the diagnosis of malnutrition and sarcopenia (1). Furthermore, the U PROBE-L6C® has been successfully employed in other study to assess the anterior rectus femoris as part of nutritional status evaluations in oncology settings (2). Another interesting study using the same device aimed to validate an AI-based system for quantifying ultrasound images of the rectus femoris muscle in patients with disease-related malnutrition. The results demonstrated strong consistency and reliability between the traditional manual measurements and the automated AI-based system (3). Here, we provide the bibliographic references that might be of interest to you.
- de Luis Roman D, García Almeida JM, Bellido Guerrero D, Guzmán Rolo G, Martín A, Primo Martín D, et al. Ultrasound Cut-Off Values for Rectus Femoris for Detecting Sarcopenia in Patients with Nutritional Risk. Nutrients [Internet]. 2024 May 21 [cited 2024 Nov 10];16(11). Available from: https://pubmed.ncbi.nlm.nih.gov/38892486/
- Palmas F, Mucarzel F, Ricart M, Lluch A, Zabalegui A, Melian J, et al. Body composition assessment with ultrasound muscle measurement: optimization through the use of semi-automated tools in colorectal cancer. Front Nutr. 2024 Apr 17;11:1372816.
- García-Herreros S, López Gómez JJ, Cebria A, Izaola O, Salvador Coloma P, Nozal S, et al. Validation of an Artificial Intelligence-Based Ultrasound Imaging System for Quantifying Muscle Architecture Parameters of the Rectus Femoris in Disease-Related Malnutrition (DRM). Nutrients [Internet]. 2024 Jun 8 [cited 2024 Dec 15];16(12):1806. Available from: https://pmc.ncbi.nlm.nih.gov/articles/PMC11206908/
Methods:
What is sample size calculus? This is a SMALL sample size!
These patients are using medicines?
There are food consumption data?
Thank you very much for your insightful comment. Indeed, the sample size is small, and this is a limitation we are aware of and have addressed in the discussion section of the manuscript. On one hand, it is worth noting that this is an ongoing study with recruitment still open, and therefore, we aim to increase the sample size. However, despite the small sample, we have obtained very interesting and statistically significant results. For this reason, the authors believe it is appropriate to publish these findings, as we consider them valuable for the scientific community. We hope that you also find them interesting.
On the other hand, we would like to inform you that the recruited patients were on different medications depending on their comorbidities (such as arterial hypertension, diabetes, dyslipidemia, etc.). Additionally, dietary intake records were collected for all participants. However, to avoid extending the length of the manuscript, these details have not been included. If you consider it appropriate and believe it could enrich the manuscript, please let us know, and we will gladly include this information.
Results:
Several parts are results description are a repetition of tables.
Following your recommendations, we have removed the results that were repeated both in the text and in the tables.
Crude logistic with small sample size is speculative!
Thank you for your comment. We understand your concern regarding the use of crude logistic regression given the small sample size. While we acknowledge that unadjusted models may be seen as speculative, the limited sample size in our study did not allow for the inclusion of multiple covariates in an adjusted model without risking overfitting or reduced statistical power. Given these constraints, we opted for a crude logistic regression analysis, which, despite its limitations, provides valuable preliminary insights into the relationships between the variables of interest. We suggest that future studies with larger sample sizes would allow for the implementation of adjusted models, thereby strengthening the validity and robustness of the findings
Why BMI is limitation? Dont make sense. Small sample size is a big limitation!
On the other hand, we completely agree with your assessment and
have therefore revised that sentence in the manuscript. Below, we provide the updated version of the sentence as it now appears in the article:
There were several limitations to the current study. First, this was a cross-sectional study with a small sample size and a low proportion of women. Multicenter trials including a larger number of patients with EGC are needed for further validation. Second, multivariate logistic regression models were not conducted, which weakened the results by excluding important confounding factors such as age, type of treatment and BMI. Third, the lack of consensus regarding cut-off values for PhA, RFCSA, RF-Y-axis and RF-AT limited the ability to compare the results with previous studies. In addition, the cross-sectional nature of the study prevented participant follow-up. Therefore, prospective studies that include patients with different types of GI cancer are essential in order to establish causal relationships derived from nutritional intervention, thereby obtaining results that can be extrapolated to the oncological population.
Once again, we would like to express our gratitude for your comments.

Reviewer 2 Report
Comments and Suggestions for Authors
This is an excellently written paper that clearly persents the aims, methods, results and conclusions.
My major remark would be that the limitations could be pointed out more clearly. First, the low sample size which is already mentioned. Second, that no multivatiate models have been conducted. As age alone is a major confounding factor, it would have been interesting if controlling for age might improve the models.
Some further remarks:
- there are some visible editing elements (strikethrough text)
Author Response
Reviewer 2
This is an excellently written paper that clearly persents the aims, methods, results and conclusions.
Thank you very much for your comment. It encourages us to continue pursuing projects like this.
My major remark would be that the limitations could be pointed out more clearly. First, the low sample size which is already mentioned.
Thank you very much for your thoughtful comment. As we mentioned to the first reviewer, the sample size is small, and this is a limitation we are aware of and have addressed in the discussion section of the manuscript. This is an ongoing study with recruitment still open, and we aim to increase the sample size. Nevertheless, despite the small sample, we have obtained very interesting and statistically significant results. For this reason, we, the authors, consider it appropriate to publish these findings, as we believe they are valuable for the scientific community. We hope you also find them interesting.
Second, that no multivariate models have been conducted. As age alone is a major confounding factor, it would have been interesting if controlling for age might improve the models.
Thank you for pointing this out. We agree with this comment. Indeed, age can be a confounding factor; however, given the current small sample size, we cannot adequately address this issue statistically at this stage.
We agree that age is a significant potential confounder and controlling for it would have been ideal in a multivariate model. However, due to the limited sample size, we were concerned that adjusting for age and other covariates would lead to overfitting or reduce the statistical power of the analysis. As a result, we chose to present univariate analyses to ensure more robust and interpretable results within the constraints of our data. We fully acknowledge that future studies with larger sample sizes would allow for the inclusion of multivariate models, which would help to account for confounders such as age and improve the accuracy of the findings.
Some further remarks:
- there are some visible editing elements (strikethrough text)
Once again, we would like to thank you for your comment. We have removed the visible editing elements from the manuscript.

Reviewer 3 Report
Comments and Suggestions for Authors
The aim of the paper is clear from the abstract. This study aims to evaluate the potential usefulness of the phase angle, the cross-sectional area of the rectus femoris and the thickness of the rectus femoris, to predict malnutrition and sarcopenia in patients with esophagogastric cancer. In the introduction we particularly appreciated the explanation of sarcopenia, which is absolutely convincing in all the aspects taken into consideration and we absolutely agree with all the methods that in the last thirty years have been taken into consideration for the evaluation of the nutritional status of a patient and described by colleagues, among other things in many clinics are still in use. The research colleagues have carried out a prospective study even if with a limited number of patients selected with an absolutely shareable criterion. A series of important studies have been conducted for the detection of nutritional status that are absolutely reproducible for the capillarity of the description. We only add that in high volume centers these types of investigations can be performed which obviously improve the outcome of the treated patients (doi.org/10.3390/jcm12072708 to be cited in the bibliography). The statistical study supports the studies carried out by colleagues, proving right the intuitions from which this paper arose. We have been dealing with artificial nutrition for many years, this is one of our first papers (PMID: 9973791 if you want you can cite it) and in fact we have never read the type of research proposed by colleagues. We agree with the limitations of the study and with the conclusions that the thickness of the rectus femoris can be the only certain data but the group of patients recruited must be expanded so as not to exclude other measurements that can be predictive, excellent iconography, good English, good bibliography
Author Response
Reviewer 3
The aim of the paper is clear from the abstract.
This study aims to evaluate the potential usefulness of the phase angle, the cross-sectional area of the rectus femoris and the thickness of the rectus femoris, to predict malnutrition and sarcopenia in patients with esophagogastric cancer.
In the introduction we particularly appreciated the explanation of sarcopenia, which is absolutely convincing in all the aspects taken into consideration and we absolutely agree with all the methods that in the last thirty years have been taken into consideration for the evaluation of the nutritional status of a patient and described by colleagues, among other things in many clinics are still in use.
The research colleagues have carried out a prospective study even if with a limited number of patients selected with an absolutely shareable criterion. A series of important studies have been conducted for the detection of nutritional status that are absolutely reproducible for the capillarity of the description.
We only add that in high volume centers these types of investigations can be performed which obviously improve the outcome of the treated patients (doi.org/10.3390/jcm12072708 to be cited in the bibliography).
The statistical study supports the studies carried out by colleagues, proving right the intuitions from which this paper arose. We have been dealing with artificial nutrition for many years, this is one of our first papers (PMID: 9973791 if you want you can cite it) and in fact we have never read the type of research proposed by colleagues. We agree with the limitations of the study and with the conclusions that the thickness of the rectus femoris can be the only certain data but the group of patients recruited must be expanded so as not to exclude other measurements that can be predictive, excellent iconography, good English, good bibliography
The authors would like to express their gratitude for your comments on the manuscript. We have incorporated the bibliographic reference you suggested into the text.

Round 2
Reviewer 1 Report
Comments and Suggestions for Authors
Again! This paper decsribes SMALL SAMPLE SIZE.
Moreover, Crude logistic with small sample size is speculative!